# Systematic Study of Reaction Conditions for Size-Controlled Synthesis of Silica Nanoparticles

**DOI:** 10.3390/nano14191561

**Published:** 2024-09-27

**Authors:** Barbara Vörös-Horváth, Ala’ Salem, Barna Kovács, Aleksandar Széchenyi, Szilárd Pál

**Affiliations:** 1Institute of Pharmaceutical Technology and Biopharmacy, Faculty of Pharmacy, University of Pécs, H-7624 Pécs, Hungary; voros-horvath.barbara@egis.hu (B.V.-H.); a.salem@kingston.ac.uk (A.S.); kovacs1@gamma.ttk.pte.hu (B.K.); szilard.pal@aok.pte.hu (S.P.); 2Quality Systems Department 3, Egis Pharmaceuticals PLC, H-1475 Budapest, Hungary; 3Department of Pharmacy, School of Life Sciences, Pharmacy and Chemistry, Kingston University, Kingston upon Thames, London KT1 2EE, UK; 4Green Chemistry Research Group, János Szentágothai Research Centre, University of Pécs, Ifjúság útja 20, H-7624 Pécs, Hungary

**Keywords:** silica nanoparticles, nanoparticle synthesis, particle size control, Stöber method, ammonium hydroxide concentration, water concentration, temperature effect

## Abstract

This study presents a reproducible and scalable method for synthesizing silica nanoparticles (SNPs) with controlled sizes below 200 nm, achieved by systematically varying three key reaction parameters: ammonium hydroxide concentration, water concentration, and temperature. SNPs with high monodispersity and controlled dimensions were produced by optimizing these factors. The results indicated a direct correlation between ammonium hydroxide concentration and particle size, while higher temperatures resulted in smaller particles with increased polydispersity. Water concentration also influenced particle size, with a quadratic relationship observed. This method provides a robust approach for tailoring SNP sizes, with significant implications for biomedical applications, particularly in drug delivery and diagnostics. Using eco-friendly solvents such as ethanol further enhances the sustainability and cost-effectiveness of the process.

## 1. Introduction

In recent years, there has been increasing interest in nanosized materials, including silica nanoparticles (SNPs). The main advantages of SNPs are that these particles are chemically inert and non-toxic in a wide concentration range [1]. Their surface can be modified with several functional groups, proteins, or enzymes [2]. Because of these advantageous properties, SNPs are widely applied in several research fields, such as drug delivery [3], diagnosis and therapy [4], catalysis [5], chemical sensing [6], and biotechnology [7].

To this day, research groups focus on developing a simple method for well-size-controlled SNP synthesis with high reproducibility, where the SNP particles are highly monodisperse and sphere-shaped. Because of the complicated reaction mechanism (monomer addition growth model [8,9] and aggregation growth model [10]) and the laborious preparation method (centrifugation and washing, dialysis, filtering), the reproducibility and size controlling of synthesis of SNPs under the 200 nm size range are not adequate. The original Stöber [11] reaction provides an excellent basis for SNP synthesis. Stöber and his colleagues prepared SNPs in the 50–900 nm size range in a comprehensive study. The reaction is a sol-gel synthesis, where the silanol precursors hydrolyze in an alcohol solvent, and the reaction is catalyzed with ammonia. They investigated the influence of solvents and different silanol precursors on particle size and shape. Several researchers have modified this method by changing the precursor concentrations [12,13] or reaction conditions, e.g., temperature [14] or reaction time [15], to synthesize SNPs with the desired size. Others work with modified Stöber reaction, the so-called seeded polymerization [16], but in this process, the effect of reactant addition rate and the number or size of seeds highly influence the product properties, which factors make it more difficult to control the reactions, with this the properties of SNPs.

We aimed to develop a simple, well-size-controlled synthesis method with high reproducibility and product yield for SNPs under 200 nm particle size. We want to focus on only the synthesis procedure, not on the reaction mechanism; it may provide practical help to researchers who deal with the application of SNPs. We assumed that with the proper choice of tetraethyl orthosilicate (TEOS), ammonium hydroxide, or water concentration and reaction temperature, we could find relationships between the SNPs’ size and reaction conditions that may be suitable to calculate the reaction conditions for synthesizing SNPs with the desired size.

We chose the reaction conditions and reagents based on literature data and our previous observations and experiences. We worked with TEOS as a silanol precursor and ethanol as a solvent. Green and co-workers described before that if the alcohol chain length increases, the first nucleation step will be slower, and the size of the nuclei will be bigger, so the final particle size will also be bigger [17]. Therefore, in our experiments, we worked with ethanol instead of methanol because if we would like to synthesize SNPs in a large amount, it is an important practical aspect that ethanol is cheaper and more eco-friendly. The choice of TEOS and ammonium hydroxide concentration is based on our previous experiments. We worked with 0.26 M TEOS concentration in all cases because the product yields were unsatisfactory at lower values. If the TEOS concentration is higher than 0.3 M, the SNP size is higher than 500 nm, and here, we focus on the 5–200 nm particle size range. We have also examined previously if the ammonium hydroxide concentration is lower than 0.0192 M, then at 0.26 M TEOS concentration particles do not form, presumably under this ammonium hydroxide concentration value the hydrolysis rate is very slow, only partially hydrolyzed monomers are formed. On the other hand, the particle size increases significantly if the ammonium hydroxide concentration is higher than 0.5 M. Based on these, we worked with a constant 0.26 M TEOS concentration and in the 0.29–0.097 M ammonium hydroxide concentration range.

Plumeré et al. [18] and J. Yang et al. [19] established that if the reaction temperature is higher, the particle size and the polydispersity index will be smaller; if the reaction temperature increases, the hydrolysis and condensation rates also increase, and at higher temperature, more secondary particles form, which leads to faster nucleation kinetics. Two research teams independently studied the formation of SNPs, and they found that because of the faster nucleation process, more nuclei are produced, so smaller and more uniform particles can be synthesized at higher reaction temperatures [14,20]. Other than this, Park et al. [21] and Kim et al. [22] determined that if the temperature is higher than 55 °C, the aggregation of nuclei dramatically increases, which will control the final particle size; hence, the particle size increases again. We considered these observations and experiences and varied the reaction temperature between 25 and 55 °C.

The water concentration also influences the hydrolysis and nucleation rates; it enhances the aggregation of nuclei, which controls the primary particle number and size. Lindberg et al. [23] and Zukoski et al. [24] observed the following trend in particle size between 0.5–17 M water concentrations: the particle size decreases if the water concentration decreases under 9 M, but over this value, the particle size increases again because of the enhanced aggregation of primary particles. High water concentration is rarely used in the Stöber synthesis because its high value limits the TEOS solubility [21], which decreases the reaction yield and lengthens the reaction time. Based on these observations, we worked in the 2–5 M water concentration range.

We characterized the SNPs using dynamic light scattering (DLS) and transmission electron microscopy (TEM), as our main focus was to tailor the size and morphology of SNPs.

## 2. Materials and Methods

### 2.1. Materials

This study used tetraethoxysilane [TEOS], (Alfa Aesar [USA], purity 98%), absolute ethanol AnalaR Normapur, purity ≥ 99.8% (VWR Chemicals, Debrecen, Hungary), 28 *w*/*w*% ammonium solution AnalaR Normapur, analytical reagent (VWR Chemicals, Debrecen, Hungary), and ultrapure water (membraPure Astacus Analytical with UV, VWR Chemicals, Debrecen, Hungary). TEOS was freshly distilled before each reaction.

### 2.2. Silica Nanoparticle Synthesis

The silica nanoparticles were prepared using hydrolysis of TEOS in ethanol in the presence of ammonium hydroxide as a catalyst. The concentration of TEOS was constant in each reaction, c = 0.26 mol/dm^3^. We systematically changed the concentration of ammonia, water, or temperature as follows:

-varied the ammonia concentration between 0.29 mol/dm^3^ and 0.097 mol/dm^3^, at a constant water concentration (5 mol/dm^3^) and a constant temperature (25 °C)-varied the water concentration between 2 mol/dm^3^ and 5 mol/dm^3^ at a constant ammonia concentration (0.29/0.194/0.097 mol/dm^3^) and a constant temperature (25 °C)-varied the temperature between 25 °C and 50 °C at a constant ammonia concentration (0.29/0.194/0.097 mol/dm^3^) and a constant water concentration (5 mol/dm^3^)

The total volume for the systematic study was 8 cm^3^. The water, ethanol, and TEOS mixture were stirred at constant temperature for 15 min. Then, the reaction mixtures were sonicated for 15 min (Bandelin Sonorex, RK 52 H, Berlin, Germany), followed by adding 10 m/m% ammonium hydroxide solution under stirring. The reaction mixtures were stirred for 24 h at a constant temperature. For all experiments, we used vials of the same size 8 or 100 cm^3^, magnetic stirring bars of the same size and shape, and a constant stirring rate (1000 min^−1^). For each synthesis, a new factory-cleaned vial and cap with PTFE membrane were used to avoid contamination with silica seeds, which can be the main cause of size heterogeneity. The synthesized silica nanoparticles were stored and found to be stable without aggregation and any change in size or morphology for several months in the residual reaction solution.

### 2.3. Characterization of Silica Nanoparticles

The size distribution, hydrodynamic diameter (d), and polydispersity index (PDI) were determined by DLS using a Malvern Zetasizer Nano S instrument. The size distribution was confirmed, and the morphology of silica nanoparticles was studied with TEM (JEM 1200 EX II and JEOL-1400 TEM [JEOL Ltd., Tokyo, Japan]). Samples for TEM experiments were prepared using drop casting “as is” samples on 400 mesh copper grids with carbon coating (Micro to Nano Ltd., Haarlem, The Netherlands). ImageJ 1.53 was used for size distribution analysis from TEM images, and the equivalent diameter for 300–400 particles was determined.

### 2.4. Statistical Analysis

Statistical analyses were conducted using Microsoft Excel^®^ 2016. Correlations between the studied parameters and the diameter of silica nanoparticles were assessed using linear regression. The coefficient of determination (R^2^) was determined to establish the fit of the models. Finally, confidence intervals for each model were examined to determine the accuracy of the prediction made by the models. All experimental results were performed in triplicates and are presented as mean ± SD.

## 3. Results

Firstly, we investigated the influence of the ammonium hydroxide concentration on the particle size if the temperature is constant, t = 25 °C. We determined that in the 0.29–0.097 M ammonium hydroxide concentration range, SNPs between 27.1 and 190.8 nm could be synthesized with high product yield (≥80%; see Table 1, A1–A3 samples). The SNPs are highly monodisperse (PDI: 0.082–0.008) and spherical (see Figure 1). The SNPs could be synthesized with high reproducibility, and three parallel syntheses have been performed.

We determined that by increasing the reaction temperature, the SNP particle size decreases at constant TEOS (0.26 M), water (5 M), and ammonium hydroxide (0.29/0.194/0.097 M) concentration. At different ammonium hydroxide concentrations, the SNP size decreased at different rates. At a 0.29 M ammonia concentration, the particle size is between 190.8 nm and 69.3 nm in the 25–55 °C temperature range (see Table 1, A1 and T30A-T55A samples). At lower an ammonia concentration, 0.194 M, the particle sizes are in the 30.2–99.1 nm interval (see Table 1, A2 and T30B-T55B samples), and at 0.097 M, these values are in the 6.1–27.1 nm size range (see Table 1, A3 and T30C-T55C samples). We performed all syntheses three times, and we found that the reproducibility of SNP synthesis is excellent; the standard deviations are very low. At each temperature and ammonium hydroxide concentration, the SNPs are highly monodisperse (PDI: 0.002–0.091) and spherical (see Figure 2). Higher PDI values (0.016–0.091) were measured at higher temperatures; we assumed this phenomenon might indicate the temperature heterogeneity in the reaction vessels. We found a linear correlation between particle size and reaction temperature, and we used it to calculate the reaction temperature for tailored particle size by regulating the temperature at constant TEOS and ammonia concentration (see Figure 3). Our results show that precise temperature control makes it possible to synthesize SNPs with a desired particle size in three different size ranges at each ammonia concentration (Figure 3 and Table 2). Statistical analysis has confirmed the model’s quality (Table 3). These calculations in the different SNP size intervals allow for SNP size tailoring or fine-tuning. At a 0.29 M ammonia concentration, we can precisely control the particle size to be between 65 and 200 nm. If we want to synthesize SNPs in the 30 and 100 nm particle size range, we should work with a constant 0.194 M ammonia concentration. At a 0.097 M ammonia concentration, SNPs under 30 nm size can be produced, and we can fine-tune the particle size from 6 nm to 23 nm.

Similarly to the size dependence of temperature, at different ammonium hydroxide concentrations, the SNP particle sizes decrease in different degrees: at a 0.29 M ammonia concentration, the particle size is between 52.18 and 190.80 nm, and there is an increase in the 2–5 M water concentration range (see Table 1 A1 and W4A-W2A samples). At a lower ammonia concentration, 0.194 M, the particle sizes are in the 27.81–99.09 nm interval (see Table 1 A2 and W4B-W2B samples), and at 0.097 M, these values are in the 6.92–27.1 nm size range (see Table 1 A3 and W4C-W2C samples, Figure 4). Here, we also found a correlation between water concentration and SNP size. By decreasing the water concentration, the SNP particle size decreased at a constant TEOS (0.26 M) and ammonia concentration (0.29/0.194/0.097 M) at 25 °C; the relation between water concentration and SNP size can be fitted with quadratic equation (Figure 5 and Figure 6). These relations also make it possible to calculate and tailor the SNP size. It is important to note that if the water concentration is lower than 3 M, the sphericity of SNPs decreases while polydispersity increases. Statistical analysis has verified the quality of the model, as shown in Table 4 and Table 5.

## 4. Discussion

The study focused on the controlled growth of various sizes of silica nanoparticles (SNPs) by systematically adjusting the ammonium hydroxide concentration, water concentration, and temperature. SNP synthesis of particles sized less than 200 nm was targeted using common approaches while ensuring high yields, narrow size distributions, and reproducibility in the synthesis of the nanoparticles. In this discussion, the consequences that the results may have will be appreciated; a comparison of the results obtained with what is already known from past work will be made, and finally, how the findings can be applied will be considered.

General rules for successfully synthesizing silica nanoparticles with the desired size are: always use freshly distilled TEOS, which tends to hydrolase and form oligomers and silica seeds at room temperature without catalysts, even with atmospheric water. Always use a new factory-cleaned vial and screw cap with a PTFE sealing membrane, as the complete removal of adsorbed silica particles from the used vial wall is only plausible by HF-containing cleaning mixtures. Magnetic stirring bars have to be cleaned with great care. Silica contaminants are major reasons for polydispersity and formation of particles with undesired size.

### 4.1. Influence of Ammonium Hydroxide Concentration

The study confirmed a strong correlation between ammonium hydroxide concentration and the size of the SNPs. Higher concentrations of ammonium hydroxide yielded larger particles, with sizes ranging from 27.1 nm to 190.8 nm within the 0.29–0.097 M concentration range at 25 °C. This trend aligns with previous findings in the literature, where the hydrolysis and condensation rates are affected by the catalyst concentration, influencing the nucleation and growth stages of particle formation. The reproducibility of the results with low standard deviations demonstrates the method’s robustness, which is crucial for scaling up the production of SNPs for industrial applications. The results also suggest that fine-tuning the ammonium hydroxide concentration can effectively control particle sizes within a targeted range, offering flexibility for different use cases, such as drug delivery systems or biosensing applications requiring specific nanoparticle sizes.

### 4.2. Influence of Temperature

The study also established that increasing the reaction temperature results in smaller SNPs across different ammonium hydroxide concentrations. The linear correlation observed between temperature and particle size provides a predictable means to tailor SNP sizes through temperature control, allowing researchers to adjust reaction conditions to achieve desired particle dimensions. This finding corroborates earlier studies, which noted that higher temperatures accelerate nucleation rates, leading to smaller and more uniform particles. However, the study also found higher temperatures increased polydispersity, especially at temperatures above 50 °C. We can assume that the temperature heterogeneity within the reaction vessel may cause uneven particle formation. Hence, while temperature control is effective for size tuning, there is a need for careful monitoring to maintain uniformity.

### 4.3. Influence of Water Concentration

Water concentration had a significant but complex influence on SNP size. The study demonstrated that decreasing water concentration generally resulted in smaller particles. However, at concentrations below 3 M, the particles tended to be less spherical and more polydisperse. This observation is consistent with existing research indicating that water concentration affects the aggregation of primary particles during the growth phase. The quadratic relationship between water concentration and particle size offers a more nuanced approach to size control than the linear temperature relationship. While this provides additional flexibility, it also necessitates careful optimization to avoid undesirable particle morphologies and polydispersity.

### 4.4. Practical Implications

The ability to precisely control SNP sizes through systematic variation of reaction conditions holds significant promise for various applications, particularly in biomedical fields. For instance, particle size can critically influence the biodistribution, cellular uptake, and clearance of nanoparticles in drug delivery. The study’s findings offer a clear framework for researchers to tailor SNPs for such applications with high reproducibility and efficiency (Stöber article). Additionally, the study’s emphasis on using eco-friendly solvents like ethanol enhances the practical utility of the method by making it more sustainable and cost-effective, especially for small and mid-scale production; this aligns with current trends in green chemistry, where minimizing environmental impact is a priority.

## 5. Conclusions

A reproducible and scalable method was successfully developed for synthesizing silica nanoparticles of sizes less than 200 nm by adjusting the ammonium hydroxide concentration, water concentration, and temperature.

The research findings indicated that ammonium hydroxide concentration possesses a linear relationship with particle size distribution, whereby increasing the concentration will lead to the formation of bigger particles. Temperature is an effective way of controlling and predicting particle size alteration linearity in which polydispersity must also be controlled at higher temperatures. Water concentration is another factor that can be used in controlling particle size, giving a parabolic dependence of particle size on the water concentration. However, the particles might not be in regular spherical shape at particularly low concentrations.

## Figures and Tables

**Figure 1 nanomaterials-14-01561-f001:**
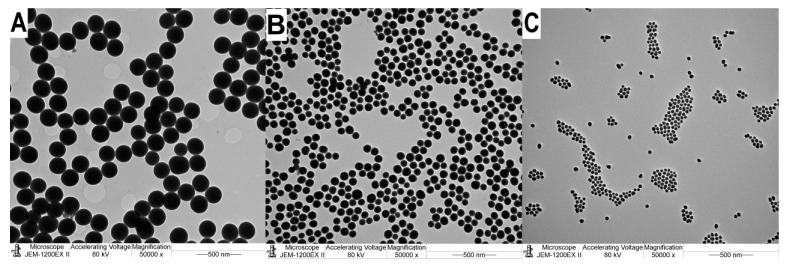
TEM images of SNPs. Samples: (**A**): A1 d = 191.38 nm, (**B**): A2 d = 91.72 nm, (**C**): A3 d = 28.33 nm. The SNPs are highly monodisperse and spherical.

**Figure 2 nanomaterials-14-01561-f002:**
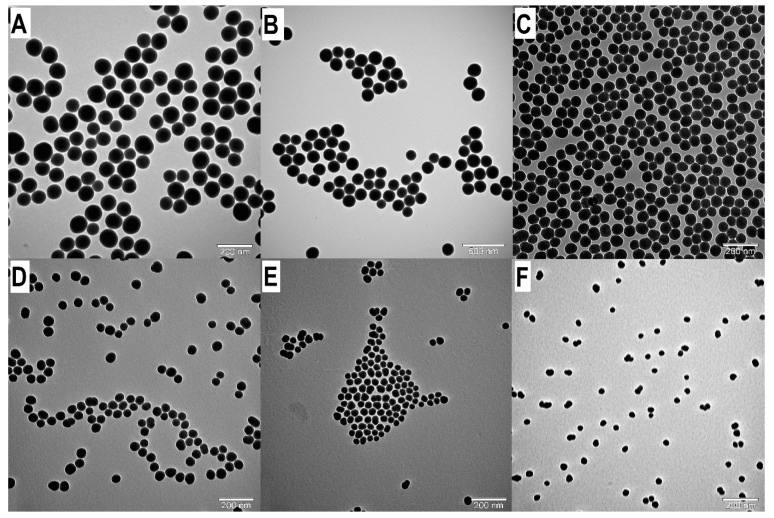
TEM images of SNPs: Examination of the influence of reaction temperature on the SNP size. (**A**): T25B d = 91.72 nm. (**B**): T30B d = 81.85 nm. (**C**): T35B d = 65.43 nm. (**D**): T40B d = 55.42 nm. (**E**): T45B d = 42.14 nm. (**F**): T50B d = 31.96 nm.

**Figure 3 nanomaterials-14-01561-f003:**
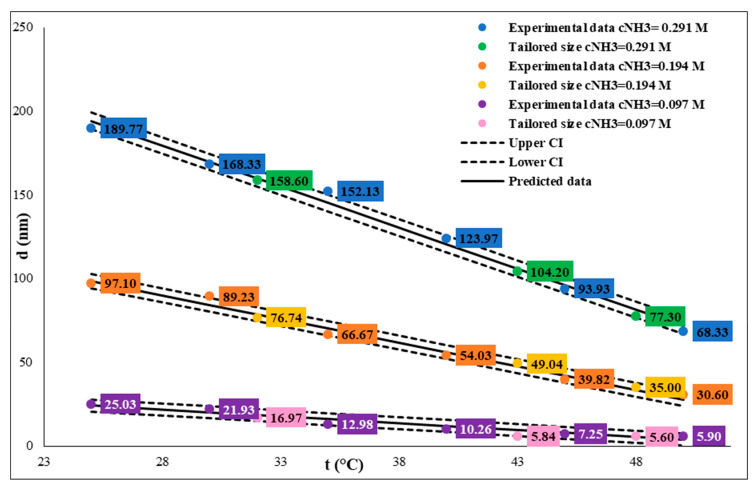
Influence of temperature on size of SNPs at different ammonium hydroxide concentrations. The TEOS concentration is constant, c = 0.26 M.

**Figure 4 nanomaterials-14-01561-f004:**
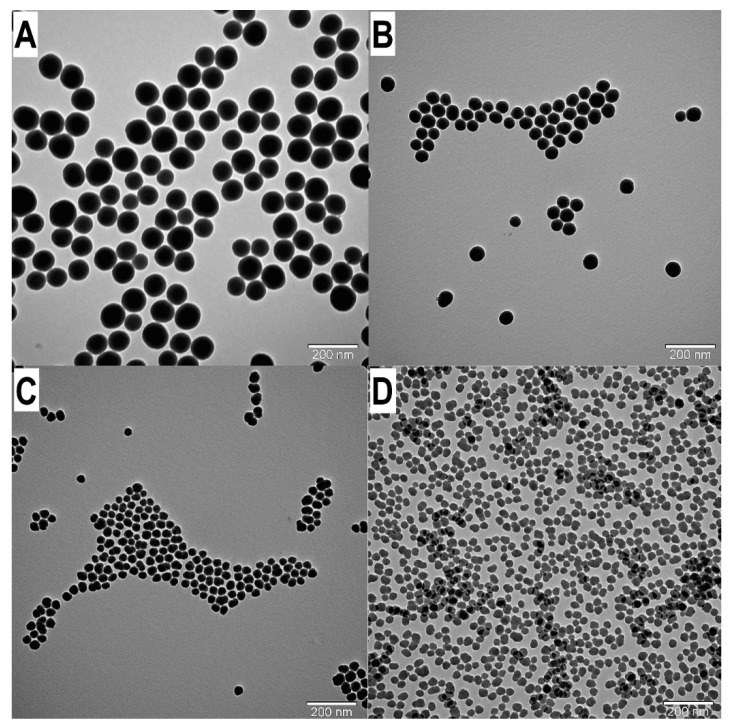
TEM images of SNPs: Examination of the influence of water concentration on the SNP size. (**A**): W5B d = 91.73 nm. (**B**): W4B d = 60.29 nm. (**C**): W3B d = 33.60 nm. (**D**): W2B d = 28.20 nm.

**Figure 5 nanomaterials-14-01561-f005:**
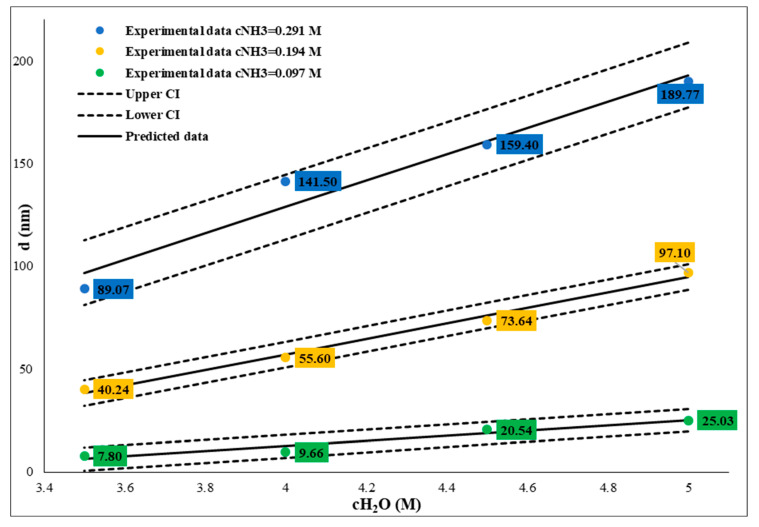
Influence of water concentration between 3.5 and 5 mol/dm^3^ on size of SNPs at different ammonium hydroxide concentrations. The TEOS concentration was constant, c = 0.26 M.

**Figure 6 nanomaterials-14-01561-f006:**
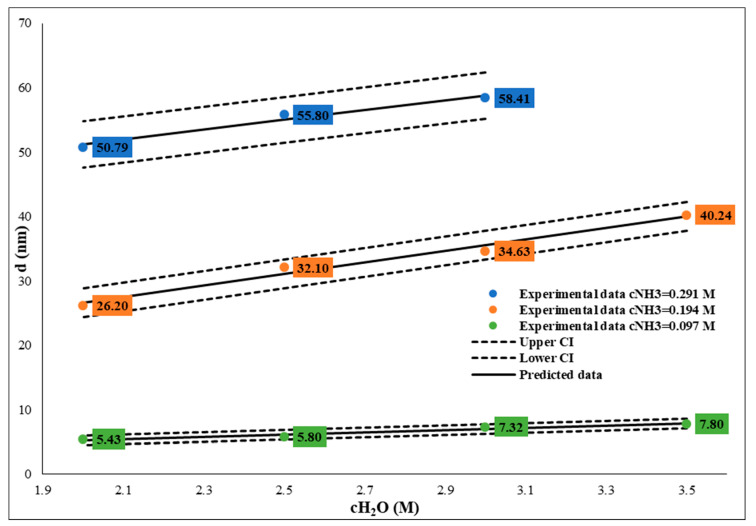
Influence of water concentration between 2 and3.5 mol/dm^3^ on size of SNPs at different ammonium hydroxide concentrations. The TEOS concentration was constant, c = 0.26 M.

**Table 1 nanomaterials-14-01561-t001:** Reaction conditions of SNP synthesis and parameters of SNPs. The TEOS concentration is constant, c = 0.26 M. Three parallel syntheses were performed.

Sample	cNH_3_ (mol/dm^3^)	cH_2_O (mol/dm^3^)	t (°C)	d_DLS_ (nm)	PDI_DLS_	d_TEM_ (nm)	PDI_TEM_
A1	0.291	5	25	190.80 ± 0.93	0.008	191.38	0.003
A2	0.194			99.09 ± 1.15	0.021	91.72	0.006
A3	0.097			27.05 ± 0.80	0.082	28.33	0.005
T30A	0.291	5	30	170.50 ± 1.17	0.002	169.75	0.001
T30B	0.194			88.60 ± 1.05	0.018	81.85	0.004
T30C	0.097			22.51 ± 0.25	0.095	23.19	0.002
T35A	0.291	5	35	153.60 ± 0.91	0.014	155.12	0.007
T35B	0.194			67.81 ± 0.77	0.005	64.53	0.015
T35C	0.097			12.08 ± 0.06	0.146	14.06	0.026
T40A	0.291	5	40	142.70 ± 1.02	0.023	145.17	0.008
T40B	0.194			56.11 ± 0.85	0.041	55.42	0.003
T40C	0.097			10.43 ± 0.12	0.226	10.99	0.018
T45A	0.291	5	45	96.35 ± 0.99	0.041	95.91	0.008
T45B	0.194			41.87 ± 0.75	0.048	42.14	0.007
T45C	0.097			7.49 ± 1.04	0.467	6.53	0.019
T50A	0.291	5	50	69.32 ± 0.88	0.016	67.80	0.012
T50B	0.194			30.15 ± 0.69	0.455	31.96	0.031
T50C	0.097			6.11 ± 0.20	0.455	8.91	0.074
W4A	0.291	4	25	142.70 ± 3.34	0.041	146.62	0.004
W4B	0.194			58.97 ± 1.58	0.043	60.29	0.002
W4C	0.097			9.54 ± 1.15	0.263	20.27	0.008
W3A	0.291	3	25	58.50 ± 2.17	0.061	54.74	0.010
W3B	0.194			34.50 ± 2.08	0.095	33.60	0.011
W3C	0.097			7.58 ± 1.64	0.190	15.71	0.006
W2A	0.291	2	25	52.18 ± 3.72	0.059	50.74	0.013
W2B	0.194			27.81 ± 8.16	0.073	28.2	0.013
W2C	0.097			6.92 ± 2.23	0.250	6.55	0.019

**Table 2 nanomaterials-14-01561-t002:** Tailoring SNP particle size with systematic change of reaction conditions. The tailored particle sizes are calculated from correlations between the experimental particle size and reaction temperature or water concentration. d_cal_ value is the calculated SNP size, d_DLS_ is the synthesized and measured SNP size.

Sample	cNH_3_ (mol/dm^3^)	cH_2_O (mol/dm^3^)	t (°C)	dcal (nm)	dDLS (nm)	PDIDLS
T32A	0.291	5	32	160.92	158.60	0.024
T32B	0.194			79.61	76.70	0.043
T32C	0.097			19.04	17.00	0.143
T43A *	0.291	5	43	106.82	104.20	0.059
T43B *	0.194			48.35	49.00	0.065
T43C *	0.097			9.52	5.80	0.070
T48A	0.291	5	48	77.3	57.33	0.004
T48B	0.194			35.0	35.00	0.093
T48C	0.097			5.19	5.60	0.245
W4.5A	0.291	4.5	25	158.44	159.40	0.028
W4.5B	0.194			78.88	73.64	0.068
W4.5C	0.097			18.01	20.54	0.090
W3.5A	0.291	3.5	25	97.99	89.07	0.026
W3.5B	0.194			46.24	40.24	0.051
W3.5C	0.097			7.51	7.80	0.176
W2.5A	0.291	2.5	25	58.44	55.78	0.057
W2.5B	0.194			30.30	32.09	0.081
W2.5C	0.097			5.30	5.82	0.227

* The syntheses were performed with a total volume of 100 cm^3^.

**Table 3 nanomaterials-14-01561-t003:** Statistical analysis of temperature–size dependence on SNP synthesis.

cNH3 (mol/dm^3^)	Linear Regression	CI	R2
0.291 M	y = −4.906x + 316.720	93%	0.9909
0.194 M	y = −2.819x + 168.630	95%	0.9846
0.0197 M	y = −0.814x + 44.412	92%	0.9326

**Table 4 nanomaterials-14-01561-t004:** Statistical analysis of a water concentration of 3.5–5 mol/dm^3^ and size dependence during SNP synthesis.

cNH_3_ (mol/dm^3^)	Linear Regression	CI	R^2^
0.291 M	y = 63.998x + 127.057	99%	0.9566
0.194 M	y = 37.222x + 96.647	95%	0.9906
0.097 M	y = 12.514x + 34.427	95%	0.9350

**Table 5 nanomaterials-14-01561-t005:** Statistical analysis of a water concentration of 2–3.5 mol/dm^3^ and size dependence during SNP synthesis.

cNH_3_ (mol/dm^3^)	Linear Regression	CI	R^2^
0.291 M	y = 7.623x + 35.940	99%	0.9679
0.194 M	y = 8.931x + 8.730	95%	0.9793
0.097 M	y = 1.726x + 1.840	95%	0.9385

## Data Availability

The raw data supporting the conclusions of this article will be made available by the corresponding author A.S. on request.

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
