# Peer review of "Systematic Study of Reaction Conditions for Size-Controlled Synthesis of Silica Nanoparticles"

_nanomaterials, 2024, doi:10.3390/nano14191561_

Round 1

Reviewer 1 Report

Comments and Suggestions for Authors

The manuscript „Systematic study of reaction conditions for size controlled synthesis of silica nanoparticles” from Vörös-Horváth et al. describes in details the parameters for the synthesis of well designed, smaller than 200 nm silica nanoparticles. The method of synthesis is not new – Stöber synthesis is well known from the 60’s – but the accurate adjustment of synthesis parameters might be still interesting for readers, especially in the nanomedicine field where narrow size distribution particles below 200 nm are often required. The manuscript fits well the scope of the journal, English language is fine.

General comments

The materials and methods section mentions that TEOS was freshly distilled before each synthesis. This might be a key step to keep low the concentration of unwanted seeds. It would deserve a  few sentences in the discussion part why this step was applied.

The article states that the synthesis is scalable, but I could not find any information about the actual reaction volume. There is information about larger than 80% yield but it is not clear what is the mass of particles or the number concentration and volume of the suspension created this way. This should be added to the article. The claim that the tailored synthesis is (up)scalable has to be supported with data or removed from the text (Page 10, line 269).

Specific comments

Page 1 line 36: check position of comma

Page 3 114 reaction mixtures were sonicated – please add the type of device used for sonication, the time and the applied energy

Page 3 line 126: please, add description, how many particles were measured for TEM analysis, what kind of diameter was measured (Feret min or max or equivalent circle…?), which software solution was used if any

Author Response

Reviewer 1

Comments and Suggestions for Authors

The manuscript "Systematic study of reaction conditions for size controlled synthesis of silica nanoparticles" from Vörös-Horváth et al. describes in details the parameters for the synthesis of well designed, smaller than 200 nm silica nanoparticles. The method of synthesis is not new – Stöber synthesis is well known from the 60's – but the accurate adjustment of synthesis parameters might be still interesting for readers, especially in the nanomedicine field where narrow size distribution particles below 200 nm are often required. The manuscript fits well the scope of the journal, English language is fine.

Answer: We greatly appreciate the insightful comments and suggestions provided by the reviewer. Your detailed feedback has enhanced the quality of our research paper. We are particularly thankful for your recommendations on unwanted seeds, which strengthened our arguments and results. Your expertise has been invaluable in refining our work, and we are grateful for the time and effort you dedicated to reviewing our manuscript.

General comments

  1. The materials and methods section mentions that TEOS was freshly distilled before each synthesis. This might be a key step to keep low the concentration of unwanted seeds. It would deserve a  few sentences in the discussion part why this step was applied.

Answer: Thank you for this valuable comment. Indeed, we have missed mentioning the reason for the fresh distillation of TEOS and some other "normal in our lab" details worth mentioning. One is that we always use new factory-cleaned vials to avoid cross-contamination with silica seeds. These pieces of information are added to chapter 1.2. Silica nanoparticle synthesis, line 120-121., and chapter3. Discussion, lines 229-235

  1. The article states that the synthesis is scalable, but I could not find any information about the actual reaction volume.

Answer: The total volume of the synthesis is added to chapter 1.2. Silica nanoparticle synthesis, line 113.

  1. There is information about larger than 80% yield but it is not clear what is the mass of particles or the number concentration and volume of the suspension created this way. This should be added to the article.

Answer: Since we kept the concentration of the TEOS constant, the mass of the formed SiO2 is also constant; since we always kept the reaction going under given conditions, we assume that the suspensions had a constant concentration of products. This claim is supported by the fact that we determined the size distribution every 30 days for several months after synthesis, and it did not change. We could not identify any TEOS in the reaction mixture after 24 h, using FTIR spectroscopy; this is why we assumed that the hydrolysis and polycondensation reaction went through stoichiometrically. We have also separated the products by centrifugation, but it was only successful for particles with a diameter of more than 100 nm, while we could not separate all the particles with a small diameter. This is why we did not give exact yield data in our manuscript; we can only claim that it is more than 80% since this was the lowest yield determined. However, we knew some products were still left in the supernatant, and the yield determination had significant errors.

  1. The claim that the tailored synthesis is (up)scalable has to be supported with data or removed from the text (Page 10, line 269).

Answer: We have missed mentioning in the manuscript that for those experiments where we have calculated the size of silica nanoparticles from linear regression data, we have made synthesis for three tailor-sized batches, two in the same size vial as for the basic experiments, and one in the 100ml of total volume for reaction mixture. The results of these experiments fit the model with the same accuracy as in the smaller batch. This was the basis of our claim of scalability.

This is now added to the manuscript in chapter 2. Results Line 193

Specific comments

  1. Page 1 line 36: check position of comma

Answer: Corrected

  1. Page 3 114 reaction mixtures were sonicated – please add the type of device used for sonication, the time and the applied energy

Answer: Data is added to chapter 1.2. Silica nanoparticle synthesis, Line 116

  1. Page 3 line 126: please, add description, how many particles were measured for TEM analysis, what kind of diameter was measured (Feret min or max or equivalent circle

…?), which software solution was used if any

Answer: ImageJ 1.53 was used for TEM image analysis, and an equivalent diameter was determined. Data is added to chapter 1.3.         Characterization of silica nanoparticles, lines 130-131

Reviewer 2 Report

Comments and Suggestions for Authors

Specific comments:

The paper indeed showed the influence of the concentration of ammonium hydroxide, water, and temperature on SNP size; however, the mechanistic basis due to which each parameter affects nanoparticle growth is not deeply explored. 

Study shows that increasing temperature increases polydispersity, it fails to explain why this is occurring or how it could be minimized. 

Life-cycle analysis or section discussion of the ecological footprint of the synthesis needs to be included in the study to justify the claims of sustainability.

It would have been important to go into a discussion of scalability for this method, regarding batch-to-batch consistency, cost analysis, and possible applications to the biomedical industry.

The paper mentions drug delivery and diagnostics as possible applications, but the study does not experimentally validate such a claim. 

Stability of the nanoparticles with long-term exposure, particularly under different physiological or environmental conditions. 

Author Response

Reviewer 2

We greatly appreciate the insightful comments and suggestions provided by the reviewer. Your detailed feedback has significantly enhanced the quality of our research paper. We are particularly thankful for your recommendations on scalability, which have strengthened our arguments and results. Your expertise has been invaluable in refining our work, and we are grateful for the time and effort you dedicated to reviewing our manuscript.

Specific comments:

  1. The paper indeed showed the influence of the concentration of ammonium hydroxide, water, and temperature on SNP size; however, the mechanistic basis due to which each parameter affects nanoparticle growth is not deeply explored.

Answer: As highlighted in the introduction, mechanistic aspects of the synthesis were not examined. We could write assumptions based on the data extracted from the literature, but this would not increase the value of our research. We focused on predicting the size for synthesis.

  1. Study shows that increasing temperature increases polydispersity, it fails to explain why this is occurring or how it could be minimized. 

Answer: As previously, we did not focus on mechanistic aspects, yet we have suggested in the manuscript that the temperature heterogeneity in the reaction vessel after the addition of catalyst may be the cause of increased size heterogeneity, but we did not do any experiment to prove this claim, we have just written down our observation.

  1. Life-cycle analysis or section discussion of the ecological footprint of the synthesis needs to be included in the study to justify the claims of sustainability.

Answer: Ethanol is generally considered a green solvent because it is derived from biological sources, not from petroleum-based chemical industries. This is why we did not explain our claim of sustainability in detail.

  1. It would have been important to go into a discussion of scalability for this method, regarding batch-to-batch consistency, cost analysis, and possible applications to the biomedical industry.

Answer: We have missed mentioning in the manuscript that for those experiments where we have calculated the size of silica nanoparticles from linear regression data, we have made synthesis for three tailor-sized batches, two in the same size vial as for the basic experiments, and one in the 100ml of total volume for reaction mixture. The results of these experiments fit the model with the same accuracy as in the smaller batch. This was the basis of our claim of scalability.This is now added to the manuscript in chapter 2. Results Line 193The scalability is proven in the laboratory scale synthesis, as we increased the basic volume of 8 ml to 100 ml with the same repeatability, as all experiments were performed as triplicates (chapter 1.4, line 135. Cost analysis is not discussed because the price of starting materials varies, which was not the aim of this study. Our claim for cost-effectiveness was intuitively made, as commercially available monodispersed silica nanoparticle suspensions have much higher prices than chemicals needed for their synthesis.

  1. The paper mentions drug delivery and diagnostics as possible applications, but the study does not experimentally validate such a claim. 

Answer: That is correct; this research did not deal with applications of silica nanoparticles, just their synthesis. We mention them in the introduction and cite some papers involved in the listed applications. As a remark, we have published many papers on this topic, but we did not mention them because we believe self-citation is unethical.

  1. Stability of the nanoparticles with long-term exposure, particularly under different physiological or environmental conditions. 

Answer: In continuation to previous answers, we did not examine long-term exposure or behavior under physiological conditions, but we have cited papers that did examine these important properties.